# Quantifying Spatiotemporal Gait Parameters with HoloLens in Healthy Adults and People with Parkinson’s Disease: Test-Retest Reliability, Concurrent Validity, and Face Validity

**DOI:** 10.3390/s20113216

**Published:** 2020-06-05

**Authors:** Daphne J. Geerse, Bert Coolen, Melvyn Roerdink

**Affiliations:** 1Department of Human Movement Sciences, Faculty of Behavioural and Movement Sciences, Vrije Universiteit Amsterdam, Amsterdam Movement Sciences, Van der Boechorststraat 7, 1081 BT Amsterdam, The Netherlands; h.coolen@vu.nl (B.C.); m.roerdink@vu.nl (M.R.); 2Department of Neurology, Leiden University Medical Center, Albinusdreef 2, 2333 ZA Leiden, The Netherlands

**Keywords:** HoloLens, spatiotemporal gait parameters, test-retest reliability, concurrent validity, face validity, healthy young adults, Parkinson’s disease

## Abstract

Microsoft’s HoloLens, a mixed-reality headset, provides, besides holograms, rich position data of the head, which can be used to quantify what the wearer is doing (e.g., walking) and to parameterize such acts (e.g., speed). The aim of the current study is to determine test-retest reliability, concurrent validity, and face validity of HoloLens 1 for quantifying spatiotemporal gait parameters. This was done in a group of 23 healthy young adults (mean age 21 years) walking at slow, comfortable, and fast speeds, as well as in a group of 24 people with Parkinson’s disease (mean age 67 years) walking at comfortable speed. Walking was concurrently measured with HoloLens 1 and a previously validated markerless reference motion-registration system. We comprehensively evaluated HoloLens 1 for parameterizing walking (i.e., walking speed, step length and cadence) in terms of test-retest reliability (i.e., consistency over repetitions) and concurrent validity (i.e., between-systems agreement), using the intraclass correlation coefficient (ICC) and Bland–Altman’s bias and limits of agreement. Test-retest reliability and between-systems agreement were excellent for walking speed (ICC ≥ 0.861), step length (ICC ≥ 0.884), and cadence (ICC ≥ 0.765), with narrower between-systems than over-repetitions limits of agreement. Face validity was demonstrated with significantly different walking speeds, step lengths and cadences over walking-speed conditions. To conclude, walking speed, step length, and cadence can be reliably and validly quantified from the position data of the wearable HoloLens 1 measurement system, not only for a broad range of speeds in healthy young adults, but also for self-selected comfortable speed in people with Parkinson’s disease.

## 1. Introduction

Mixed-reality headsets, such as Microsoft’s HoloLens (Microsoft Corporation, Redmond, WA, USA; Figure 1A), are untethered and consist of a non-occluding holographic display unit through which 3D holograms can be anchored in the wearer’s environment. Mixed-reality has been studied in various applied contexts, including, but not limited to, assistance during surgical interventions (e.g., [1,2]), remote medical assessments or support (e.g., [3,4]), and walking-ability assessments with holographic obstacle avoidance [5]. To ensure that holographic content realistically blends with the real environment, the headset must sense its position and orientation with respect to the surrounding, which requires many sensors, including an inertial measurement unit, four ‘environment-understanding’ cameras, a depth camera (see for example Hübner et al. [6] for a HoloLens sensor evaluation study) and a set of algorithms (Simultaneous Localization and Mapping, SLAM) to compute the position and orientation of the headset with respect to its surrounding, while at the same time mapping the structure of that environment. These position and orientation streams of the headset in itself provide a rich source of information, for example to quantify what the wearer is doing (e.g., sitting, standing, turning, walking) and to parameterize such acts (e.g., speed, step length, cadence for walking). Studies utilizing the position and orientation data of HoloLens 1 have already revealed good accuracy for foot-step detection [7] and excellent agreement with manual stopwatch scores for completion times of several functional mobility tasks in healthy young and older adults [3].

The aim of the current study is to comprehensively evaluate HoloLens 1 for quantifying the spatiotemporal gait parameters of walking (i.e., walking speed, step length and cadence) from the position data by determining test-retest reliability (i.e., what are the limits of agreement of derived spatiotemporal gait parameters over repetitions?) and concurrent validity (i.e., how well do the derived spatiotemporal gait parameters agree with those determined concurrently with a reference motion-registration system?). We do this with a group of healthy young adults walking at slow, comfortable, and fast speeds (to also be able to examine face validity in terms of differences across conditions, i.e., do the derived spatiotemporal gait parameters vary in to-be-expected directions?) as well as in a group of people with Parkinson’s disease walking at their self-selected comfortable speed. We were particularly interested in this group, as we are currently examining the potential efficacy and usability of HoloLens for alleviating freezing of gait in people with Parkinson’s disease through patient-tailored 2D and 3D holographic cues, such as bars on the ground to step over (Figure 1B; for more details: Netherlands Trial Register, Trial NL7523). However, to date, it is not known if (Parkinsonian) gait can be parameterized reliably and with good concurrent validity with the HoloLens 1, which seems a prerequisite for presenting holographic cues in a patient-tailored (i.e., inter-cue distances matching one’s gait pattern) and intelligent or assist-as-needed manner (i.e., only receiving cues when deviating from a reference gait pattern [8]). We expect that spatiotemporal gait parameters quantified with the HoloLens are reliable for test-retest, agree well with those derived from a reference motion-registration system (in fact, narrower limits of agreement are expected for between-systems differences than for within-system repetitions), with well-detectable differences between instructed walking-speed conditions in to-be-expected directions.

## 2. Materials and Methods

### 2.1. Subjects

A convenience sample of 23 healthy young adults (mean (range): age 21.0 (18–28) years, Montreal Cognitive Assessment score 28.3 (26–30), 5 males and 18 females) and 24 people with Parkinson’s disease (age 67.0 (55–76) years, Montreal Cognitive Assessment score 25.6 (12–30), Movement Disorder Society version of the Unified Parkinson’s Disease Rating Scale motor score 40.0 (15–59), Hoehn and Yahr stage 2.2 (2–3), 15 males and 9 females) participated in this study. Subjects were recruited at the Leiden University Medical Center (i.e., healthy young adults were students/employees, people with Parkinson’s disease via outpatient clinic). Subjects had to be 18 years or older. Healthy young adults had to have normal cognitive function (Montreal Cognitive Assessment score (0–30) ≥ 26; [9]) and (corrected to) normal vision. People with Parkinson’s disease had to be able to walk independently, meet the UK Parkinson’s Disease Society Brain Bank clinical diagnostic criteria [10], have a Hoehn and Yahr stage of 1–4 [11] and experience freezing of gait while on their medication. Candidate subjects were not included if they had (additional) neurological or orthopedic diseases interfering with gait. All subjects provided written informed consent, and the study was approved by the local Medical Ethical Committee (P18.065).

### 2.2. Experimental Set-Up and Procedure

We measured walking concurrently with HoloLens 1 (Figure 1A) and the Interactive Walkway (i.e., the reference motion-registration system [12,13]; Figure 1C). The HoloLens provides position and orientation data of the head in the environment in three directions, which was recorded using custom-written software at a sampling frequency of approximately 30 Hz. The position data were used to derive spatiotemporal gait parameters. The Interactive Walkway was considered the reference motion-registration system for collecting walking data. It is an instrumented 8-m walkway for markerless full-body 3D motion registration. The Interactive Walkway has been validated against a marker-based motion-registration system for gait parameters [12] and outcome measures of walking adaptability [14] in a group of healthy adults. In addition, it has been used in people with stroke [15] and Parkinson’s disease [13], demonstrating good known-groups validity with age-matched controls for a range of outcome measures, including walking speed, step length and cadence. Whereas marker-based motion-registration systems have long been considered the gold standard for gait analyses [16,17,18], the use of markerless systems increases because they provide accurate and reliable data [19,20,21,22] efficiently and without requiring bodily contact (i.e., no time is lost in marker placement and calibration, physical contact is not needed, safe distance to the participant can be easily maintained). The Interactive Walkway uses four spatially and temporally integrated Kinect v2 sensors to obtain the 3D position of 25 body points, such as head, spine shoulder and feet. The Interactive Walkway set-up of this study was the same as the validated set-up described in Geerse et al. [13]. A projector (EPSON EB-585W, ultra-short-throw 3LCD projector) was used to present the 8-m walkway. Interactive Walkway data were sampled at approximately 30 Hz using custom-written software utilizing the Kinect for Windows Software Development Kit (SDK 2.0).

At the beginning of the experiment, subjects had a familiarization period of walking with the HoloLens. Subsequently for healthy young adults, overground walking data was collected over a wide range of walking speeds. Healthy young adults were instructed to walk over the 8-m walkway at a slow walking speed (‘Strolling, like in a shopping mall or on a boulevard.’; SWS), comfortable walking speed (‘A walking speed that you can maintain for a longer period of time.’; CWS), and fast walking speed (‘Walk as fast as possible, without running.’; FWS) while wearing the HoloLens. All conditions were repeated twice and performed in a fixed order (2× comfortable, 2× slow, 2× fast), totaling six trials. People with Parkinson’s disease were measured while they were on their medication. After the familiarization period of walking with the HoloLens, people with Parkinson’s disease walked twice the 8-m walkway at their comfortable walking speed while wearing the HoloLens.

### 2.3. Data Pre-Processing and Analysis

Walking data were first pre-processed per system separately. The Interactive Walkway provides the position of 25 body points in three directions, of which the spine shoulder, spine base, and ankles were used to calculate the spatiotemporal gait parameters (Table 1). Pre-processing steps for Interactive Walkway data were identical to Geerse et al. [13] and included combining the data of the four Kinects, interpolating to ensure a constant sampling frequency, removing outliers from the data and interpolating with a spline algorithm. The HoloLens provides the position of the head in three directions. HoloLens’ position data were linearly interpolated using the timestamps provided by the custom-written software to ensure a constant sampling frequency of 30 Hz. Then, the coordinate system of the HoloLens was aligned to that of the Interactive Walkway using a transformation matrix, which was calculated for each trial over the entire 8-m walkway. Preprocessed and aligned time series of both systems in the anterior-posterior and vertical direction (Figure 2) for the central 6 m of the walkway were then used to quantify walking speed, step length, and cadence, as detailed in Table 1, on a per-walk basis. The raw and preprocessed time series of both systems are available as supplementary material (Appendix A).

### 2.4. Statistical Analysis

Test-retest reliability (i.e., between repetitions for all conditions, separately for both systems) and concurrent validity (i.e., between-systems agreement for all conditions and repetitions) were evaluated with: 1) the intraclass correlation coefficient for absolute agreement (ICC_(A,1)_; [24]; ICCs above 0.60 and 0.75 represented good and excellent agreement, respectively [25]) and 2) the bias (trial 1-trial 2 or Interactive Walkway-HoloLens) and the limits of agreement (LoA) obtained with a Bland-Altman analysis [26].

Face validity of HoloLens 1 data for parameterizing walking was evaluated through the between-conditions validity by comparing walking speed, step length and cadence across the three walking-speed conditions for both repetitions with a repeated-measures ANOVA with the factor Condition (SWS, CWS, FWS); with increasing instructed walking speeds, performed walking speeds, step lengths and cadences were expected to increase. The assumption of sphericity was checked according to Girden [27]. If Greenhouse–Geisser’s epsilon exceeded 0.75, the Huynh-Feldt correction was applied; otherwise the Greenhouse–Geisser correction was used. Main effects were examined with a least significant difference post-hoc test. Effect sizes were quantified with *η_p_*^2^.

## 3. Results

One healthy young adult was excluded from the analyses due to drift in the HoloLens data of the CWS condition, probably due to loss of tracking of the system. In addition, one person with Parkinson’s disease was excluded from the analyses, since no data were recorded with the HoloLens.

### 3.1. Test-Retest Reliability

Test-retest reliability for walking speed, step length and cadence can be found in Table 2. The ICCs indicated an excellent test-retest reliability for both systems for healthy young adults (Interactive Walkway: ICC_(A,1)_ ≥ 0.863, HoloLens: ICC_(A,1)_ ≥ 0.861; Table 2) and people with Parkinson’s disease (Interactive Walkway: ICC_(A,1)_ ≥ 0.765, HoloLens: ICC_(A,1)_ ≥ 0.778; Table 2). LoA were comparable between systems and overall quite wide. Small but significant biases between repetitions were found for walking speed, step length, and cadence, but only for CWS of healthy young adults, for both systems alike (*p* ≤ 0.005).

### 3.2. Concurrent Validity

An excellent between-systems agreement was observed for walking speed, step length and cadence, with all ICC_(A,1)_ ≥ 0.928 (Table 3, in which the results of trial 1 are presented). Correspondingly, the LoA were quite narrow (LoA in Table 3), especially relative to those seen over repetitions (LoA in Table 2). Still, walking speed, step length and to a lesser extent cadence derived from HoloLens data generally slightly but significantly underestimated those of the Interactive Walkway, with biases increasing with faster instructed walking speeds and for healthy young adults (Table 3). Similar results were observed for trial 2: excellent ICC-values, narrow LoA with overall small but significant biases.

### 3.3. Face Validity

A significant effect of instructed walking-speed conditions (3 levels: SWS, CWS, FWS) was found for performed walking speed (*F*(1.8,38.8) = 396.7, *p* < 0.001, *η_p_*^2^ = 0.950), step length (*F*(1.4,29.8) = 213.2, *p* < 0.001, *η_p_*^2^ = 0.910), and cadence (*F*(1.8,37.9) = 264.3, *p* < 0.001, *η_p_*^2^ = 0.926). Walking speed, step length and cadence all increased with instructed walking speeds (Table 2; Figure 3; data presented are for trial 1), and differed significantly between all levels (*p* < 0.001). The same results were found for trial 2: walking speed, step length, and cadence all changed significantly in to-be-expected directions.

## 4. Discussion

The aim of the current study was to determine test-retest reliability, concurrent validity, and face validity of HoloLens 1 data for quantifying spatiotemporal gait parameters in healthy young adults and people with Parkinson’s disease. In line with what we expected, walking speed, step length and cadence quantified with HoloLens 1 data were reliable for test-retest, agreed well with those concurrently derived with a reference motion-registration system, and differed significantly between walking-speed conditions in to-be-expected directions, demonstrating face validity. This comprehensive evaluation of HoloLens 1 for parameterizing walking enabled us to put the reliability and validity results into perspective. Below, we point-wise discuss: (1) within-system repeatability in relation to between-systems reproducibility, (2) within-system repeatability in relation to between-conditions differences, and (3) between-systems biases in relation to between-conditions differences.

First, and notwithstanding the observation that ICCs were both excellent over repetitions and between systems, we found that the between-systems reproducibility was better than the within-system test-retest repeatabilities. This was evidenced by ICCs closer to 1 and considerably narrower LoA between systems (Table 3) than over repetitions within systems (Table 2). The observation that the between-systems variation was considerably lower than the within-system(s) variation over repetitions is encouraging in that walking speeds, step lengths and cadences may be determined interchangeably with HoloLens 1 and Interactive Walkway. The advantage of the HoloLens in this regard is that the HoloLens is a wearable system that could potentially be used for various applications in free-living environments, such as patient-tailored assist-as-needed holographic cueing in people with Parkinson’s disease (Figure 1B). This was the first study examining the ability of the HoloLens 1 to quantify walking in a clinical population and it was not a priori known if Parkinsonian gait could be parameterized reliably and with good concurrent validity. Results proved to be excellent for both test-retest reliability and concurrent validity, with comparable, if not better, scores for people with Parkinson’s disease than for the healthy young adults who do not suffer from gait abnormalities. Still, the relatively poorer within-system repeatability (which could further degrade when assessed over sessions instead of within sessions as in the current study), for both systems and groups alike, may potentially hinder the detection of differences in walking speed, step length and cadence between groups or over conditions.

This brings us to the second point. Our face validity findings revealed that the relatively poorer test-retest repeatability did not demonstrably hinder the detection of between-conditions differences using HoloLens 1 data, at least not in the context of the present comparisons. We observed differences between the three instructed walking-speed conditions in healthy young adults (Figure 3), in to-be-expected directions, that is, faster walking speeds are associated with larger steps and higher cadences [12,28]. Moreover, these between-conditions results were comparable over repetitions. Hence, the relatively larger variability seen over repetitions (than between systems) did not negate the discriminative ability of walking speed, step length and cadence derived from HoloLens 1 data for between-conditions differences, which demonstrates face validity. Face validity can also be demonstrated by comparing groups that are known to walk differently (also known as known-groups validity). Sun et al. [3] already evaluated the discriminative ability of functional mobility assessment outcomes derived from HoloLens 1 data. However, no significant differences were found between healthy young and older adults, probably because their walking ability was not too dissimilar for the assessed tasks. In the current study, we can compare the spatiotemporal gait parameters derived with HoloLens position data between people with Parkinson’ disease and healthy young adults walking at their comfortable walking speed. People with Parkinson’s disease walked with significantly lower walking speeds (−11.7 cm/s; *t*(43) = 2.09, *p* = 0.042), smaller step lengths (−10.2 cm; *t*(36.4) = 3.60, *p* = 0.001), and higher cadences (5.8 steps/min; *t*(43) = −2.42, *p* = 0.020) than healthy young adults. Although these groups were not age-matched, results were in expected directions. People with Parkinson’s disease typically walk with a reduced step length and walking speed compared to age-matched healthy controls, while cadence is often preserved or slightly faster in people with Parkinson’s disease suffering from freezing of gait [13,20,28,29,30].

Thirdly, we discuss the between-systems biases in relation to the face validity findings. Small but systematic biases were observed for walking speed and step length, which increased in magnitude with walking speed and for healthy young adults (Table 3). The origin of these biases probably resides in between-systems differences in the cumulative distance traveled in the anterior-posterior direction along the walkway, which was used to calculate walking speed and step length (Table 1), based on time indices of the Interactive Walkway between the 1- and 7-m lines of the walkway. Differences between systems for this cumulative distance ranged between 15 and 18 cm (2.5%–3.0%) for healthy young adults and between –5.5 and 15.4 cm (−0.9%–2.6%) for people with Parkinson’s disease, which by and large corresponded to the percentage bias found for walking speed (0.7%–3.2% of reference values) and step length (1.2%–3.3%). Note that the percentage bias for cadence was considerably smaller (0.4%–1.4%). Quantifying spatial gait parameters from HoloLens 1 position data critically depends on the accuracy of the underlying (real-time) mapping of the direct environment and the simultaneous localization of itself within this map. Although, recently, the mapping accuracy of HoloLens 1 was favorably evaluated by Hübner et al. [6], large-scale drift in tracking and deviations in the maps caused by it were also noted, which may explain the deviations in cumulative distance in the current study, as well as the between-systems biases in derived walking speeds and step lengths. When we put these between-systems biases in perspective, we see that the observed between-systems biases in walking speed and step length are considerably smaller (at least four times) than the differences between walking-speed conditions (i.e., compare biases in Table 3 to Figure 3), as well as between the two groups (at least two times).

## 5. Conclusions

Spatiotemporal gait parameters derived from the position data of HoloLens 1 were reliable for test-retest, agreed well with those derived from a reference motion-registration system, and differed between walking-speed conditions and groups in to-be-expected directions. In particular, walking speed, step length, and cadence can be reliably and validly quantified from HoloLens 1 position data, not only in healthy young adults over a broad range of speeds, but also in people with Parkinson’s disease walking at their self-selected comfortable walking speed, indicating that the HoloLens 1 may also be used as a wearable measurement system in selected clinical populations.

## Figures and Tables

**Figure 1 sensors-20-03216-f001:**
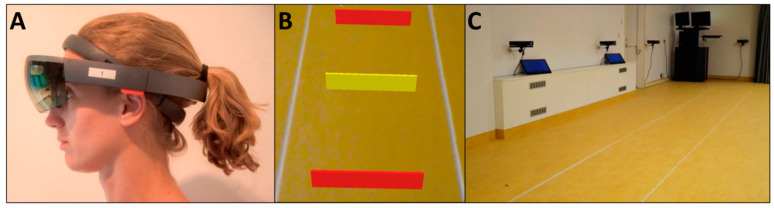
HoloLens 1 (**A**), 3D holographic cues seen through HoloLens 1 (**B**) and Interactive Walkway used as a reference motion-registration system (**C**).

**Figure 2 sensors-20-03216-f002:**
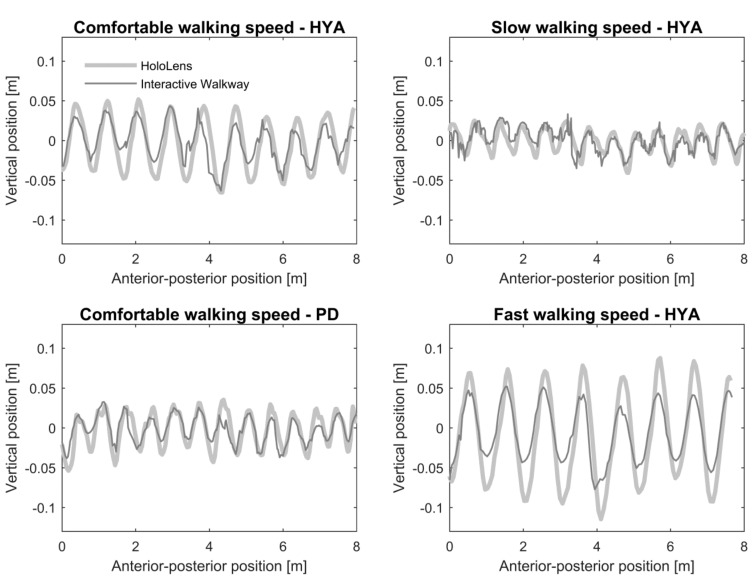
Examples of the position in the anterior-posterior (*x*-axis) and vertical (*y*-axis) direction for HoloLens’ head data and Interactive Walkway spine shoulder data during slow, comfortable and fast walking speed trials of a healthy young adult (HYA) and for self-selected comfortable walking speed of a person with Parkinson’s disease (PD). The mean was subtracted from the time series in vertical direction for visualization purposes.

**Figure 3 sensors-20-03216-f003:**
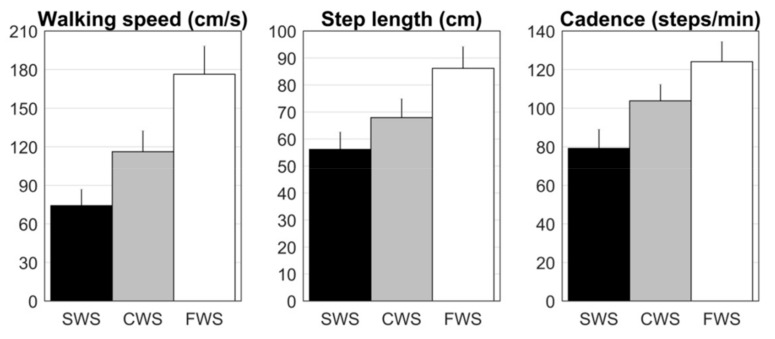
Mean values and between-subjects standard deviations for walking speed, step length and cadence derived from HoloLens data for healthy young adults walking at an instructed slow (SWS, black bars), comfortable (CWS, gray bars) and fast walking speed (FWS, white bars). All spatiotemporal gait parameters differed between walking-speed conditions in to-be-expected directions, demonstrating face validity.

**Table 1 sensors-20-03216-t001:** Calculation methods of spatiotemporal gait parameters using HoloLens and Interactive Walkway position data.

	HoloLens	Interactive Walkway
**Walking speed** **(cm/s)**	The distance travelled in the anterior-posterior direction between the 1-m and 7-m lines on the walkway divided by the time, using the position of the HoloLens (i.e., head position).	The distance travelled in the anterior-posterior direction between the 1-m and 7-m lines on the walkway divided by the time, using the data of the spine shoulder.
**Step length** **(cm)**	The mean of the differences in the anterior-posterior direction of consecutive step locations. Time estimates of step locations were defined as the maxima of the head in the vertical direction. Minimal time between maxima was set at 0.6 times the step duration, which was calculated using the highest frequency found in the time series in vertical direction or 0.5 times the highest frequency found in the time series in mediolateral direction in case of small displacements in the vertical direction (mainly for people with Parkinson’s disease).	The median of the differences in the anterior-posterior direction of consecutive step locations. Step locations were determined as the median anterior–posterior position of the ankles during the single-support phase (i.e., between foot off and foot contact of the contralateral foot; [12,13]). Estimates of foot contact and foot off were defined as the maxima and minima of the anterior–posterior time series of the ankles relative to that of the spine base [12,13,23].
**Cadence** **(steps/min)**	Calculated from the number of steps in the time interval between the first and last estimate of foot contact. Estimates of foot contact were defined as the minima of the head in the vertical direction. Minimal time between minima was set at 0.6 times the step duration, which was calculated using the highest frequency found in the time series in vertical direction or 0.5 times the highest frequency found in the time series in mediolateral direction in case of small displacements in the vertical direction (mainly for people with Parkinson’s disease).	Calculated from the number of steps in the time interval between the first and last estimate of foot contact. Estimates of foot contact were defined as the maxima of the anterior-posterior time series of the ankles relative to that of the spine base [12,13,23].

**Table 2 sensors-20-03216-t002:** Test-retest reliability for spatiotemporal gait parameters of instructed slow walking speed (SWS), comfortable walking speed (CWS) and fast walking speed (FWS) conditions in healthy young adults (HYA) and CWS in people with Parkinson’s disease (PD) derived from Interactive Walkway and HoloLens data.

			Interactive Walkway					HoloLens				
			Trial 1	Trial 2				Trial 1	Trial 2			
			Mean ± SD	Mean ± SD	Bias (95% LoA)	*t*-Statistics	ICC_(A,1)_	Mean ± SD	Mean ± SD	Bias (95% LoA)	*t*-Statistics	ICC_(A,1)_
**Walking speed** **(cm/s)**	**HYA**	**SWS**	76.1 ± 13.0	78.1 ± 12.5	−2.0 (−14.8 10.8)	*t*(21) = −1.42, *p* = 0.169	0.863	74.2 ± 12.6	76.2 ± 12.2	−2.0 (−14.6 10.5)	*t*(21) = −1.49, *p* = 0.152	0.861
		**CWS**	119.6 ± 16.8	125.8 ± 17.1	−6.2 (−19.0 6.6)	*t*(21) = −4.45, *p* < 0.001 *	0.871	116.2 ± 16.2	122.1 ± 16.4	−5.9 (−18.4 6.6)	*t*(21) = −4.33, *p* < 0.001 *	0.870
		**FWS**	182.3 ± 22.9	180.4 ± 23.7	1.9 (−14.9 18.8)	*t*(21) = 1.04, *p* = 0.310	0.932	176.3 ± 21.8	174.7 ± 22.8	1.7 (−14.7 18.1)	*t*(21) = 0.93, *p* = 0.362	0.930
	**PD**	**CWS**	105.2 ± 21.4	106.9 ± 21.4	−1.7 (−16.8 13.3)	*t*(22) = −1.09, *p* = 0.289	0.935	104.5 ± 20.7	106.1 ± 20.8	−1.6 (−16.2 13.1)	*t*(22) = −1.01, *p* = 0.324	0.935
**Step length** **(cm)**	**HYA**	**SWS**	57.5 ± 6.6	58.0 ± 5.6	−0.5 (−5.9 4.9)	*t*(21) = −0.83, *p* = 0.417	0.899	56.2 ± 6.4	56.8 ± 5.4	−0.6 (−6.2 5.0)	*t*(21) = −0.95, *p* = 0.355	0.884
		**CWS**	69.6 ± 7.1	71.2 ± 6.9	−1.5 (−6.0 3.0)	*t*(21) = −3.12, *p* = 0.005 *	0.928	67.9 ± 7.0	69.7 ± 6.6	−1.8 (−6.4 2.8)	*t*(21) = −3.64, *p* = 0.002 *	0.911
		**FWS**	89.1 ± 8.2	88.6 ± 9.0	0.5 (−5.1 6.2)	*t*(21) = 0.84, *p* = 0.411	0.945	86.2 ± 8.0	85.5 ± 8.3	0.7 (−5.4 6.7)	*t*(21) = 0.99, *p* = 0.334	0.928
	**PD**	**CWS**	58.4 ± 11.8	58.7 ± 11.4	−0.3 (−8.2 7.7)	*t*(22) = −0.34, *p* = 0.739	0.941	57.8 ± 11.5	57.8 ± 10.9	0.0 (−7.9 7.8)	*t*(22) = −0.03, *p* = 0.977	0.939
**Cadence** **(steps/min)**	**HYA**	**SWS**	79.4 ± 10.2	80.9 ± 9.9	−1.4 (−9.9 7.0)	*t*(21) = −1.57, *p* = 0.132	0.903	79.2 ± 9.8	80.6 ± 9.6	−1.4 (−9.6 6.8)	*t*(21) = −1.58, *p* = 0.130	0.903
		**CWS**	104.3 ± 8.5	107.4 ± 8.8	−3.1 (−8.6 2.4)	*t*(21) = −5.20, *p* < 0.001 *	0.892	103.8 ± 8.5	106.5 ± 8.7	−2.7 (−9.0 3.6)	*t*(21) = −3.90, *p* < 0.001 *	0.890
		**FWS**	125.7 ± 11.5	125.9 ± 10.3	−0.2 (−9.8 9.4)	*t*(21) = −0.17, *p* = 0.867	0.903	124.1 ± 10.4	123.7 ± 9.8	0.4 (−5.7 6.5)	*t*(21) = 0.60, *p* = 0.553	0.953
	**PD**	**CWS**	110.3 ± 7.7	111.2 ± 7.3	−0.9 (−11.1 9.2)	*t*(22) = −0.87, *p* = 0.391	0.765	109.6 ± 7.6	110.5 ± 7.0	−0.9 (−10.4 8.7)	*t*(22) = −0.87, *p* = 0.392	0.778

Abbreviations: SD = between-subjects standard deviation; LoA = limits of agreement; ICC_(A,1)_ = intraclass correlation coefficient for absolute agreement. * Significant over-repetitions difference (*p* < 0.05).

**Table 3 sensors-20-03216-t003:** Concurrent validity (i.e., between-systems agreement) for spatiotemporal gait parameters of instructed slow walking speed (SWS), comfortable walking speed (CWS) and fast walking speed (FWS) conditions in healthy young adults (HYA) and CWS in people with Parkinson’s disease (PD), presented for trial 1.

			Interactive Walkway	HoloLens			
			Mean ± SD	Mean ± SD	Bias (95% LoA)	*t*-Statistics	ICC_(A,1)_
**Walking speed**	**HYA**	**SWS**	76.1 ± 13.0	74.2 ± 12.6	1.9 (0.9 2.9)	*t*(21) = 17.63, *p* < 0.001 *	0.988
**(cm/s)**		**CWS**	119.6 ± 16.8	116.2 ± 16.2	3.4 (1.9 5.0)	*t*(21) = 20.36, *p* < 0.001 *	0.978
		**FWS**	182.3 ± 22.9	176.3 ± 21.8	6.0 (3.2 8.8)	*t*(21) = 19.58, *p* < 0.001 *	0.963
	**PD**	**CWS**	105.2 ± 21.4	104.5 ± 20.7	0.7 (−2.0 3.3)	*t*(22) = 2.32, *p* = 0.030 *	0.998
**Step length**	**HYA**	**SWS**	57.5 ± 6.6	56.2 ± 6.4	1.3 (−0.2 2.7)	*t*(21) = 9.08, *p* < 0.001 *	0.973
**(cm)**		**CWS**	69.6 ± 7.1	67.9 ± 7.0	1.7 (−0.8 4.2)	*t*(21) = 6.18, *p* < 0.001 *	0.956
		**FWS**	89.1 ± 8.2	86.2 ± 8.0	2.9 (0.4 5.4)	*t*(21) = 10.82, *p* < 0.001 *	0.928
	**PD**	**CWS**	58.4 ± 11.8	57.8 ± 11.5	0.6 (−2.0 3.3)	*t*(22) = 2.17, *p* = 0.041 *	0.992
**Cadence**	**HYA**	**SWS**	79.4 ± 10.2	79.2 ± 9.8	0.3 (−1.4 2.0)	*t*(21) = 1.61, *p* = 0.123	0.996
**(steps/min)**		**CWS**	104.3 ± 8.5	103.8 ± 8.5	0.4 (−1.8 2.7)	*t*(21) = 1.73, *p* = 0.097	0.990
		**FWS**	125.7 ± 11.5	124.1 ± 10.4	1.6 (−3.5 6.7)	*t*(21) = 2.89, *p* = 0.009 *	0.963
	**PD**	**CWS**	110.3 ± 7.7	109.6 ± 7.6	0.7 (−1.1 2.6)	*t*(22) = 3.75, *p* = 0.001*	0.988

Abbreviations: SD = between-subjects standard deviation; LoA = limits of agreement; ICC_(A,1)_ = intraclass correlation coefficient for absolute agreement. * Significant between-systems difference (*p* < 0.05).

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
