# Peer review of "Quantifying Spatiotemporal Gait Parameters with HoloLens in Healthy Adults and People with Parkinson’s Disease: Test-Retest Reliability, Concurrent Validity, and Face Validity"

_sensors, 2020, doi:10.3390/s20113216_

Round 1
Reviewer 1 Report
===================
GENERAL COMMENTS
===================
The main objective of this study is to determine test-retest reliability and concurrent validity of HoloLens 1 for quantifying spatiotemporal gait parameters. The authors have already published recent articles on the use of the Hololens system in gait analysis.
This work is very relevant with important clinical implications. Indeed, spatiotemporal gait parameters derived from the position data of HoloLens 1 seems to be reliable for test-retest. Also, this system could be a new way to assess the gait capabilities in neurological populations. This paper is well-structured and written.
However, I have some concerns about several points developed below that I would like the authors to consider in their revised version.
===================
SPECIFIC COMMENTS
===================
MAJOR COMMENTS
----------------
First, the choice of populations is not well enough argued. Why did you choose Parkinson’s disease among neurological disease? Is the gait pattern of these patients of particular interest for this study?
Then, authors analyze the known-groups validity by comparing the CWS walking speed, step length and cadence between healthy young adults and people with Parkinson’s disease, but the interest of this comparison is not clearly demonstrated in the introduction. They find difference between groups, but this result was expected and does not provide any benefits. It's surprising that they didn't recruit age-matched participants.
Please show the relevance of the face validity in terms of known-groups difference or delete it.
Second, why didn't Parkinson's patients realize the two others walking conditions (slow and fast walking speed) ? It would have been interesting to have access to these data since it is possible that the reliability and validity are less under these conditions specifically in this neurological population.
Third, the principle of reliability studies is to compare a new system with a gold-standard system. I am not convinced that the Interactive Walkway used by authors can be considered as a gold standard. This system is not well known and is based on few publications by this research team.
----------------
MINOR COMMENTS
----------------
L53-L59. This part would be more suitable in the Method.
L88. It’s not necessary to assess cognitive performance for young adults.
Results part: Please choice between p < 0.05 or p = …
Author Response
Reviewer 1
GENERAL COMMENTS
The main objective of this study is to determine test-retest reliability and concurrent validity of HoloLens 1 for quantifying spatiotemporal gait parameters. The authors have already published recent articles on the use of the Hololens system in gait analysis.
This work is very relevant with important clinical implications. Indeed, spatiotemporal gait parameters derived from the position data of HoloLens 1 seems to be reliable for test-retest. Also, this system could be a new way to assess the gait capabilities in neurological populations. This paper is well-structured and written.
However, I have some concerns about several points developed below that I would like the authors to consider in their revised version.
SPECIFIC COMMENTS
MAJOR COMMENTS
Comment 1
First, the choice of populations is not well enough argued. Why did you choose Parkinson’s disease among neurological disease? Is the gait pattern of these patients of particular interest for this study?
Response: The reviewer rightly pointed out that we have not specifically argued why we included a group of people with Parkinson’s disease. This study is part of Project Holocue funded by the Michael J. Fox Foundation in which we examine the potential efficacy and usability of holographic cues for alleviating ‘freezing of gait’ in people with Parkinson’s disease in free-living environments. ‘Freezing of gait’ is defined as an episodic inability (lasting seconds) to generate effective stepping. There is a general consensus among clinicians that cues, such as stripes on the ground to step onto, can be an effective therapeutic to help alleviate ‘freezing of gait’ episodes once they have occurred. Whereas recent studies suggest that 3D cues may be more effective than 2D cues, patients’ responses to specific cueing modalities vary strongly, calling for an individually tailored approach for presenting the right type(s) of cues. With the Holocue-application, designed for the HoloLens, we therefore aim to alleviate ‘freezing of gait’ by presenting patient-tailored holographic visual cues, such as horizontal bars to step over. More details about the project can be found in the Netherlands Trial Register (Trial NL7523).
The main advantage of the HoloLens is that is can be used as an assistive device, as in Project Holocue, but at the same time as a measurement device. Gait parameters derived from the position data of the HoloLens, which can be determined reliably and validly as demonstrated in this paper, can be used to customize the distance between the cues and to measure the effect of the cues on the gait pattern of the user, without requiring additional measurement devices. In addition, the cues could also be presented only when experiencing ‘freezing’, circumventing side effect such as developing cue dependency, being more fatiguing and being more attention demanding. Intelligent or assist-as-needed cueing, only receiving cues when deviating from a reference gait pattern, appeared most successful in maintaining steady gait (Ginis et al., 2017). For this, valid information of the gait pattern is needed, which explains the need for a validation study in people with Parkinson’s disease.
We have added the following sentences to the Introduction to shortly explain our motives: “We were particularly interested in this group, as we are currently examining the potential efficacy and usability of HoloLens for alleviating freezing of gait in people with Parkinson’s disease through patient-tailored 2D and 3D holographic cues, such as bars on the ground to step over (Figure 1B; for more details: Netherlands Trial Register, Trial NL7523). However, to date, it is not known if (Parkinsonian) gait can be parameterized reliably and with good concurrent validity with the HoloLens 1, which seems a prerequisite for presenting holographic cues in a patient-tailored (i.e., intercue-distances matching one’s gait pattern) and intelligent or assist-as-needed manner (i.e., only receiving cues when deviating from a reference gait pattern; [8]).”
Ginis, P.; Heremans, E.; Ferrari, A.; Dockx, K.; Canning, C.G.; Nieuwboer, A. Prolonged Walking with a Wearable System Providing Intelligent Auditory Input in People with Parkinson's Disease. Front Neurol 2017, 8, 128.
Comment 2
Then, authors analyze the known-groups validity by comparing the CWS walking speed, step length and cadence between healthy young adults and people with Parkinson’s disease, but the interest of this comparison is not clearly demonstrated in the introduction. They find difference between groups, but this result was expected and does not provide any benefits. It's surprising that they didn't recruit age-matched participants. Please show the relevance of the face validity in terms of known-groups difference or delete it.
Response: Although in principle face validity in terms of known-groups differences can also be established for non-age-matched groups, we have given the between-groups comparison a less prominent position in the paper as the differences may have been amplified a bit given the age difference (but see below). We have removed the known-groups validity from the Introduction, Methods and Results sections and have now only briefly mentioned the between-groups results in the Discussion section (“Face validity can also be demonstrated by comparing groups that are known to walk differently […], while cadence is often preserved or slightly faster in people with Parkinson’s disease suffering from freezing of gait [13,20,28-30].”).
Note that comparable results were to be expected with age-matched healthy participants. In one of our previous studies (Geerse et al., 2018), we have compared gait parameters derived with the Interactive Walkway between people with Parkinson’s disease experiencing ‘freezing of gait’ and age-matched controls. No significant differences were found for cadence (-0.2 steps/min), but people with Parkinson’s disease had significantly lower walking speeds (-22.6 cm/s) and smaller step lengths (-11.7 cm) than age-matched controls. Note that differences between healthy young adults and the current groups of people with Parkinson’s disease were smaller, except for cadence.
Geerse, D.J.; Roerdink, M.; Marinus, J.; van Hilten, J.J. Assessing walking adaptability in Parkinson’s disease: “the Interactive Walkway”. Front Neurol 2018, 9, 1096.
Comment 3
Second, why didn't Parkinson's patients realize the two others walking conditions (slow and fast walking speed) ? It would have been interesting to have access to these data since it is possible that the reliability and validity are less under these conditions specifically in this neurological population.
Response: That is certainly true, but we don’t have the data. As already mentioned, this study was part of a larger project. The laboratory session for people with Parkinson’s disease consisted –besides this between-system and between-trials validation- of several other walking trials aimed at eliciting freezing of gait. That is, they were asked to complete various ‘freezing’-provoking tasks and routes multiple times to assess ‘freezing of gait’ in the lab, under multiple conditions, namely walking without HoloLens (baseline condition), walking with the HoloLens without Holocue functionality (to examine possible distractor effects of wearing an unfamiliar device) and walking with the Holocue-application (to examine the immediate effect on freezing of gait occurrences and durations). Since walking in people with Parkinson’s disease is quite demanding, and measurements already took up to 2 hours (including questionnaires), we decided not to include more validation trials to limit the load for the participants.
Comment 4
Third, the principle of reliability studies is to compare a new system with a gold-standard system. I am not convinced that the Interactive Walkway used by authors can be considered as a gold standard. This system is not well known and is based on few publications by this research team.
Response: We agree with the reviewer that the employed system is not widely known, and that there are just a few studies showing that gait parameters can be determined interchangeably with more established marker-based motion-registration systems. As a consequence, considering the Interactive Walkway as a gold standard is indeed quite a step. We have therefore replaced the term ‘gold-standard motion-registration system’ with the term ‘reference motion-registration system’ throughout the paper.
Note that markerless systems are increasingly being implemented because they provide good-quality data in an efficient manner without requiring bodily contact. Moreover, the techniques behind these systems continue to improve over time, as supported by many favorable validation studies against marker-based motion-registration systems (Dolatabadi et al., 2016; Eltoukhy et al., 2017; Müller et al., 2017; Springer & Yogev Seligmann, 2016). We have briefly mentioned this in the Methods section (“Whereas marker-based motion-registration systems have long been considered the gold standard for gait analyses [16-18], the use of markerless systems increases because they provide accurate and reliable data [19-22] efficiently and without requiring bodily contact (i.e., no time is lost in marker placement and calibration, physical contact is not needed, safe distance to the participant can be easily maintained).”).
Dolatabadi, E.; Taati, B.; Mihailidis, A. Concurrent validity of the Microsoft Kinect for Windows v2 for measuring spatiotemporal gait parameters. Med Eng Phys 2016, 38(9). 952-958.
Eltoukhy, M.; Kuenze, C.; Oh, J.; Jacopetti, M.; Wooten, S.; Signorile, J. Microsoft Kinect can distinguish differences in over-ground gait between older persons with and without Parkinson's disease. Med Eng Phys 2017, 44, 1-7.
Müller, B.; Ilg, W.; Giese, M.A.; Ludolph, N. Validation of enhanced kinect sensor based motion capturing for gait assessment. PLoS One 2017, 12(4), e0175813.
Springer, S.; Yogev Seligmann, G. Validity of the Kinect for Gait Assessment: A Focused Review. Sensors (Basel) 2016, 16(2), 194.
MINOR COMMENTS
Comment 5
L53-L59. This part would be more suitable in the Method.
Response: We have moved some parts to the final paragraph of the Introduction explaining the aims of this study.
Comment 6
L88. It’s not necessary to assess cognitive performance for young adults.
Response: In order to make sure that participants understood the instructions given, we have added a minimal score of 26 on the Montreal Cognitive Assessment as inclusion criteria for healthy adults (see Netherlands Trial Register, Trial NL7523). We have therefore not removed this information from the Methods section.
Comment 7
Results part: Please choice between p < 0.05 or p = …
Response: We were not sure what the reviewer meant with this comment. We have only used < or > when we reported multiple values or when values were lower than 0.001 (p<0.001). When only one value was reported, we used the =-sign to indicate the exact p-value. This was only the case for the analysis of the known-groups validity, which is now presented in the Discussion section.
Reviewer 2 Report
Authors tested a methodology for the calculation of spatio-temporal parameters related to gait. Parameters were computed by using head movement recorded by HoloLens system. Performance of the methodology was compared with a markerless system. Reliability and validity were also assessed. Tests were performed on both healthy subjects and PD patients.
The paper shows an innovative approach. However, I have major concerns that authors should consider for improve the scientific rigor and impact of the manuscript.
- Although the proposed is innovative, it is not clear the real application of such methodology. Which are the advantages of use this approach with respect the common methods to compute spatiotemporal parameters both indoor (optoelectronic systems) and outdoor (inertial sensors)? Authors should stress this point in the introduction to make more explicit the scientific impact of the paper.
- Introduction lacks important aspects, as also demonstrated by the low number of references. For example some sentences on the best practice to measure spatio-temporal parameters should be included, as well the use of markerless system.
- To consider markerless system as gold standard motion capture system is a hypothesis quite unrealistic. Authors should consider, at least, to discuss this point among the limitation of the study.
- Since the orientation of the head represents the feature for parameter extraction, authors should explain if they calibrated the sensors embedded in the HoloLens and in which way (see for example, https://doi.org/10.1115/1.2355685). An incorrect alignment of the sensor can lead to incorrect estimation of parameters considering the methods reported in Table 1.
- More details on Interactive Walkway performance should be reported, also using previous studies in literature.
- Authors should report that the test-retest repeatability is only used for quantifying the reliability among strides of the same task and two consecutive repetitions, no intra-day and inter-day analyses were performed.
- Lines 289-296. I suggest to put them into a separate Conclusion section.
Author Response
Reviewer 2
Authors tested a methodology for the calculation of spatio-temporal parameters related to gait. Parameters were computed by using head movement recorded by HoloLens system. Performance of the methodology was compared with a markerless system. Reliability and validity were also assessed. Tests were performed on both healthy subjects and PD patients.
The paper shows an innovative approach. However, I have major concerns that authors should consider for improve the scientific rigor and impact of the manuscript.
Comment 1
Although the proposed is innovative, it is not clear the real application of such methodology. Which are the advantages of use this approach with respect the common methods to compute spatiotemporal parameters both indoor (optoelectronic systems) and outdoor (inertial sensors)? Authors should stress this point in the introduction to make more explicit the scientific impact of the paper.
Response: The main advantage is motion registration in a real-time mapped environment, into which in real-time holographic content can be stably presented. One can envision ample applications that this new technology would enable. We have developed the Holocue application, providing 2D or 3D holographic cues to help alleviate ‘freezing of gait’ in people with Parkinson’s disease in their free-living environments. ‘Freezing of gait’ is defined as an episodic inability (lasting seconds) to generate effective stepping. There is a general consensus among clinicians that cues, such as stripes on the ground to step onto, can be an effective therapeutic to help alleviate ‘freezing of gait’ episodes once they have occurred. Whereas recent studies suggest that 3D cues may be more effective than 2D cues, patients’ responses to specific cueing modalities vary strongly, calling for an individually tailored approach for presenting the right type(s) of cues. With the Holocue-application, designed for the HoloLens, we therefore aim to alleviate ‘freezing of gait’ by presenting patient-tailored holographic visual cues, such as horizontal bars to step over. More details about the project can be found in the Netherlands Trial Register (Trial NL7523).
The main advantage of the HoloLens is that is can be used as an assistive device, as with the Holocue-application, but at the same time as a measurement device. Gait parameters derived from the position data of the HoloLens, which can be determined reliably and validly as demonstrated in this paper, can be used to customize the distance between the cues and to measure the effect of the cues on the gait pattern of the user, without requiring additional measurement devices. In addition, the cues could also be presented only when experiencing ‘freezing’, circumventing side effect such as developing cue dependency, being more fatiguing and being more attention demanding. Intelligent or assist-as-needed cueing, only receiving cues when deviating from a reference gait pattern, appeared most successful in maintaining steady gait (Ginis et al., 2017). For this, valid information of the gait pattern is needed, which explains the need for a validation study.
We have explained this in short in the Introduction, including a picture of holographic content (Figure 1B): “We were particularly interested in this group, as we are currently examining the potential efficacy and usability of HoloLens for alleviating freezing of gait in people with Parkinson’s disease through patient-tailored 2D and 3D holographic cues, such as bars on the ground to step over (Figure 1B; for more details: Netherlands Trial Register, Trial NL7523). However, to date, it is not known if (Parkinsonian) gait can be parameterized reliably and with good concurrent validity with the HoloLens 1, which seems a prerequisite for presenting holographic cues in a patient-tailored (i.e., intercue-distances matching one’s gait pattern) and intelligent or assist-as-needed manner (i.e., only receiving cues when deviating from a reference gait pattern; [8]).”
Ginis, P.; Heremans, E.; Ferrari, A.; Dockx, K.; Canning, C.G.; Nieuwboer, A. Prolonged Walking with a Wearable System Providing Intelligent Auditory Input in People with Parkinson's Disease. Front Neurol 2017, 8, 128.
Comment 2
Introduction lacks important aspects, as also demonstrated by the low number of references. For example some sentences on the best practice to measure spatio-temporal parameters should be included, as well the use of markerless system.
Response: Point well taken, although we have added the suggested aspects to the Methods section to not break the flow of the Introduction: “Whereas marker-based motion-registration systems have long been considered the gold standard for gait analyses [16-18], the use of markerless systems increases because they provide accurate and reliable data [19-22] efficiently and without requiring bodily contact (i.e., no time is lost in marker placement and calibration, physical contact is not needed, safe distance to the participant can be easily maintained).”.
Comment 3
To consider markerless system as gold standard motion capture system is a hypothesis quite unrealistic. Authors should consider, at least, to discuss this point among the limitation of the study.
Response: We agree with the reviewer that the employed markerless Interactive Walkway system is not widely known, and that there are just a few studies showing that gait parameters can be determined interchangeably with more established marker-based motion-registration systems. As a consequence, considering the Interactive Walkway as a gold standard is indeed quite a step. We have therefore replaced the term ‘gold-standard motion-registration system’ with the term ‘reference motion-registration system’ throughout the paper.
Note that markerless systems are increasingly being implemented because they provide good-quality data in an efficient manner without requiring bodily contact. Moreover, the techniques behind these systems continue to improve over time, as supported by many favorable validation studies against marker-based motion-registration systems (Dolatabadi et al., 2016; Eltoukhy et al., 2017; Müller et al., 2017; Springer & Yogev Seligmann, 2016).
Dolatabadi, E.; Taati, B.; Mihailidis, A. Concurrent validity of the Microsoft Kinect for Windows v2 for measuring spatiotemporal gait parameters. Med Eng Phys 2016, 38(9). 952-958.
Eltoukhy, M.; Kuenze, C.; Oh, J.; Jacopetti, M.; Wooten, S.; Signorile, J. Microsoft Kinect can distinguish differences in over-ground gait between older persons with and without Parkinson's disease. Med Eng Phys 2017, 44, 1-7.
Müller, B.; Ilg, W.; Giese, M.A.; Ludolph, N. Validation of enhanced kinect sensor based motion capturing for gait assessment. PLoS One 2017, 12(4), e0175813.
Springer, S.; Yogev Seligmann, G. Validity of the Kinect for Gait Assessment: A Focused Review. Sensors (Basel) 2016, 16(2), 194.
Comment 4
Since the orientation of the head represents the feature for parameter extraction, authors should explain if they calibrated the sensors embedded in the HoloLens and in which way (see for example, https://doi.org/10.1115/1.2355685). An incorrect alignment of the sensor can lead to incorrect estimation of parameters considering the methods reported in Table 1.
Response: In the study mentioned by the reviewer, one type of sensor was used, namely accelerometers. Therefore, a linear compensation procedure of the drift was conducted. The HoloLens is a ‘holographic’ computer equipped with an inertial measurement unit (IMU, which includes an accelerometer, gyroscope and a magnetometer), four ‘environment-understanding’ cameras, and a depth camera (Kinect v3 sensor). These sensors are factory installed and pre-calibrated. In addition, the HoloLens uses a set of algorithms collectively called Simultaneous Localization and Mapping (SLAM) to compute the position and orientation of the headset with respect to its surrounding, while at the same time mapping the structure of that environment (Cadena et al., 2016), a feature also used nowadays in driverless cars, drones, Mars rover navigation and even inside the human body (Cadena et al., 2016; Omar et al., 2018; Guillaume et al., 2017; García et al.,2016; Leng et al., 2013; Mountney et al., 2006). In this way, the camera systems can compensate for numerical drift in the integration of accelerometer data, and the IMU (with an accelerometer) can handle camera occlusions. The output of this sensor fusion is a stream of position and orientation data of the headset, and at the same time a spatial map of the world.
In the context of mixed reality, the quality of the position and orientation of the headset are of utmost importance, since a holographic object should be stably presented in the real environment such that it maintains its location when the HoloLens changes position and orientation (i.e., when the wearer moves its head, or walks around the hologram), otherwise the illusion of blending digital content with the real world does not work. Given that holograms can be stably presented in the real environment, and are also perceived as such by the HoloLens wearers (see e.g., Coolen et al. 2020), it implies that the position and orientation data are thus of high quality, with solid sensor alignment and fusion under the hood.
Cadena, C.; Carlone, L.; Carrillo, H.; Latif, Y.; Scaramuzza, D.; Neira, J.; Reid, I.; Leonard, J.J. Past, present, andfutureofsimultaneouslocalizationandmapping: Towardstherobust-perceptionage. IEEETrans. Robot. 2016, 32, 1309–1332.
Omar Takleh, T.T.; Bakar, N.A.; Rahman, S.; Hamzah, R.; Abd Aziz, Z. A brief survey on SLAM methods in autonomous vehicles. Int. J. Eng. Technol. 2018, 7, 38–43.
Guillaume, B.; Zayed, A.; Li, Y.; Sébastien, G. Simultaneous localization and mapping: A survey of current trends in autonomous driving. IEEE Trans. Intell. Veh. 2017, 2, 194–220.
García,S.;Guillén,M.;Barea,R.;Bergasa,L.;Molinos,E.IndoorSLAMformicroaerialvehiclescontrolusing monocular camera and sensor fusion. In Proceedings of the 2016 International Conference on Autonomous Robot Systems and Competitions (ICARSC), Braganca, Portugal, 4–6 May 2016; pp. 205–210.
Leng, C.; Cao, Q.; He, M.; Huang, Z. Development of a Mars Rover with mapping and localization system. Res. J. App. Sci. Eng. Technol. 2013, 6, 2127–2130.
Mountney, P.; Stoyanov, D.; Davison, A.; Yang, G.Z. Simultaneous stereoscope localization and soft-tissue mapping for minimal invasive surgery. In Medical Image Computing and Computer-Assisted Intervention—MICCAI 2006; Larsen, R., Nielsen, M., Sporring, J., Eds.; Springer: Berlin/Heidelberg, Germany, 2006; pp. 347–354.
Coolen, B.; Beek, P.J.; Geerse, D.J.; Roerdink, M. Avoiding 3D obstacles in mixed reality: does it differ from negotiating real obstacles? Sensors (Basel) 2020, 20(4), E1095.
Comment 5
More details on Interactive Walkway performance should be reported, also using previous studies in literature.
Response: We have added the requested detailed information about the Interactive Walkway performance (by referring to recent validation studies) to the Methods section: “The Interactive Walkway has been validated against a marker-based motion-registration system for gait parameters [12] and outcome measures of walking adaptability [14] in a group of healthy adults. In addition, it has been used in people with stroke [15] and Parkinson’s disease [13], demonstrating good known-groups validity with age-matched controls for a range of outcome measures, including walking speed, step length and cadence.”.
Comment 6
Authors should report that the test-retest repeatability is only used for quantifying the reliability among strides of the same task and two consecutive repetitions, no intra-day and inter-day analyses were performed.
Response: We have mentioned this in the Discussion section (“[…] (which could further degrade when assessed over sessions instead of within sessions as in the current study).”). One may expect that the variation over days may become even larger than the variations seen over repetitions, resulting in lower within-system repeatabilities (i.e., for both systems alike). However, there is no reason to expect that the within-system reliability would differ between systems so that differences over time can be measured interchangeably between systems (considering the excellent concurrent validity).
Comment 7
Lines 289-296. I suggest to put them into a separate Conclusion section.
Response: Good suggestion, we have transferred this paragraph to a separate Conclusion section.
Round 2
Reviewer 1 Report
The manuscript has been improved and responses to my questions.
I have no further question.
Reviewer 2 Report
My only concern remain the system used as reference, however paper is well written and authors answered to my previous doubts.